# Automatic Blob Detection Method for Cancerous Lesions in Unsupervised Breast Histology Images

**DOI:** 10.3390/bioengineering12040364

**Published:** 2025-03-31

**Authors:** Vincent Majanga, Ernest Mnkandla, Zenghui Wang, Donatien Koulla Moulla

**Affiliations:** Department of Computer Science, University of South Africa, Preller Street, Muckleneuk Ridge, Pretoria 1709, South Africa; majanvi@unisa.ac.za (V.M.); wangz@unisa.ac.za (Z.W.)

**Keywords:** augmentation, stain normalization, connected components analysis, active contours, segmentation, blob detection, deep learning

## Abstract

The early detection of cancerous lesions is a challenging task given the cancer biology and the variability in tissue characteristics, thus rendering medical image analysis tedious and time-inefficient. In the past, conventional computer-aided diagnosis (CAD) and detection methods have heavily relied on the visual inspection of medical images, which is ineffective, particularly for large and visible cancerous lesions in such images. Additionally, conventional methods face challenges in analyzing objects in large images due to overlapping/intersecting objects and the inability to resolve their image boundaries/edges. Nevertheless, the early detection of breast cancer lesions is a key determinant for diagnosis and treatment. In this study, we present a deep learning-based technique for breast cancer lesion detection, namely blob detection, which automatically detects hidden and inaccessible cancerous lesions in unsupervised human breast histology images. Initially, this approach prepares and pre-processes data through various augmentation methods to increase the dataset size. Secondly, a stain normalization technique is applied to the augmented images to separate nucleus features from tissue structures. Thirdly, morphology operation techniques, namely erosion, dilation, opening, and a distance transform, are used to enhance the images by highlighting foreground and background pixels while removing overlapping regions from the highlighted nucleus objects in the image. Subsequently, image segmentation is handled via the connected components method, which groups highlighted pixel components with similar intensity values and assigns them to their relevant labeled components (binary masks). These binary masks are then used in the active contours method for further segmentation by highlighting the boundaries/edges of ROIs. Finally, a deep learning recurrent neural network (RNN) model automatically detects and extracts cancerous lesions and their edges from the histology images via the blob detection method. This proposed approach utilizes the capabilities of both the connected components method and the active contours method to resolve the limitations of blob detection. This detection method is evaluated on 27,249 unsupervised, augmented human breast cancer histology dataset images, and it shows a significant evaluation result in the form of a 98.82% F1 accuracy score.

## 1. Introduction

Nuclear detection is a key step for most CAD systems targeting image analysis, such as in automated capturing and grading for breast cancer (BC) tissue samples. The early diagnosis of BC relies heavily on how cancerous lesions spread across nuclei and tissue glands in an image. Conventional methods require medical practitioners to use the Bloom-Richardson grading system to determine the grade and extent of tumor cells morphing into normal nucleic cells, the degree of morphing, and the extent to which the tumor is increasing [1]. Therefore, the grading system directly correlates with the shape and appearance of breast cancer nucleus objects in histology images.

Recently, there has been an increased demand for the early detection of breast cancer (BC) at screening sites/hospitals, thus opening up avenues for new research. The early, automatic detection of cancer increases the chances of making accurate decisions for successful treatment. Therefore, screening procedures are analyzed through computer-aided (CAD) systems, which use medical images to improve the clinical efficiency and confidentiality. Conventionally, the evaluation of medical images has been quite time-inefficient and varies from person to person; thus, most of the recent research studies have targeted the analysis of medical images to aid in clinical diagnosis [2].

Automated nucleus object detection has also been challenging given the large number of nucleus objects, the sizes of high-resolution digitized medical images, and the variable sizes, appearances, textures, and shapes of individual nucleus objects. The hematoxylin and eosin (H&E) staining procedure has shown significant results, being the preferred standard for the histologic examination of human glands/tissues [3]. The early diagnosis of BC relies heavily on the extent to which cancerous lesions spread in histology images—specifically in nuclei and tissue glands—thus assisting in prognosis.

The distinction brought about by BC classification as either malignant or benign has led researchers to extensively explore the application of deep learning methods in the assessment of progression and treatment in cancer. Histopathology represents one use case that exemplifies the application of deep learning methods to big image data given their size and complexity. DL techniques have shown large success in various fields, namely object detection, image recognition, and classification. CAD systems targeting cancer detection in histology images have various research prospects. This study focuses on examining the tissue characteristics, cell nucleus isolation/identification, and cancerous lesion detection, thus assisting early diagnosis.

The detection of cancerous lesions in breast histology images is solely dependent on the segmentation of nucleus objects. Most of these segmentation methods largely revolve around techniques that target regions of interest (ROI) and edges/boundaries, namely watershed segmentation, active contours, and other grouped techniques that involve different morphology operations [4].

However, segmentation techniques suffer from oversegmentation and thus do not work well for overlapping nucleus cells. Active contours have been increasingly used in image segmentation; however, their major limitation is the inability to deal with the object inhomogeneity in large images, which leads to the segmentation of multiple objects as a single object [5].

In histopathology, the morphological appearance of different features and structures in an image, such as nucleus cells or glands, often indicates the presence of a disease. In the case of BC, the shape and morphological characteristics of nuclei in histology images correlate with the disease aggressiveness [6]. Conventionally, thresholding and morphology operations are the preferred techniques for image segmentation. Morphology operations, as proposed by [7], are used to pre-process, threshold, and further post-process images to detect edges. Automated segmentation techniques such as grayscaling, median filtering, and bottom–top hat filters, as proposed by [8], use pre-processing steps to enhance the image contrast. Thresholding is used to identify regions of interest, while post-processing morphology techniques, namely dilation, area opening, and hole filling, are used to improve the final segmentation results.

Thresholding and binary morphological operations are also presented in [9]; they utilize dilation and erosion to identify the breast region of interest (ROI), masking and isolating it from unwanted pectoral muscle regions. In [10], image pre-processing is handled through normalization, segmentation through color de-convolution for nucleus enhancement, and data augmentation to increase the dataset size, and a binary threshold is used to detect nucleus edges in the images.

Hence, CAD systems are important in predicting BC via accurately and efficiently isolating and identifying the locations of nucleus objects and segmenting them so that relevant morphological features related to BC may be obtained and used for subsequent detection.

The segmentation of cancerous nucleus cells in breast histology images presents several challenges, as discussed herein; thus, this study proposes an automatic blob detection method for cancerous lesions in unsupervised breast histology images.

This proposed method pre-processes images through various augmentation methods, namely random cropping, rotation, vertical and horizontal shift translation, and scaling adjustments to increase the size of the dataset. The H&E stain normalization technique is applied to the resultant augmented images to remove color inconsistencies and separate and isolate nucleic features from tissue structures. Morphology operations, namely erosion, dilation, opening, and a distance transform, are then used to highlight foreground and background pixels in the image.

Subsequently, the connected components analysis method is introduced to group highlighted pixel components with similar characteristics and assign them their relevant labeled components (binary masks). The active contours method then utilizes the resultant binary masks for further segmentation by resolving the inhomogeneity of ROI boundaries/edges. Lastly, a deep learning recurrent neural network (RNN) model automatically detects and extracts nucleus objects that contain cancerous lesions and their edges in the histology images via the blob detection method. This proposed approach utilizes the capabilities of the connected components analysis method and active contours method to resolve the limitations of blob detection.

The main contributions of this paper are as follows.

Augmentation methods are used to deal with data scarcity. Additionally, stain normalization is used to deal with color inconsistencies.Morphology operations enhance the image by highlighting important features. The connected components analysis method is used to group components with similar characteristics and assist in separating overlapping and non-overlapping objects.The active contours method uses the obtained binary masks from the connected components analysis to highlight and isolate the edges/boundaries of ROIs. Further, the blob detection method is used to resolve undersegmentation from the previous step and identify BC lesions (blobs) from the previously obtained masked images.

The rest of this paper is as follows: we discuss existing related work in Section 2, the proposed segmentation method is described in Section 3, the results and discussion are given in Section 4, the conclusions are presented in Section 5, and the future work is described in Section 6.

## 2. Related Work

The utilization of CAD systems in pathology can extensively increase the efficiency and precision of pathologist’s choices, thus being beneficial to patients. With new advancements in the image processing sphere, several methods have emerged and have been suggested for the accurate detection of BC. In various studies, the automatic segmentation and classification of nucleus cells emerges as a repetitive task, being especially challenging for histology images. The segmentation of nuclei/histological structures in histology images is challenging given that the most microscopic parts of the cells are unpredictable and show rare visual angles [11].

A review of some of the most preferred CAD techniques for the analysis of BC from histology images has been provided in [12]. The authors delve into the different types of image staining procedures commonly used, the various methods of histopathology analysis, and the various parameters used for the performance evaluation of classifiers; they also compare different algorithms used for nucleus detection, segmentation, and classification and discuss future possibilities.

The authors of [13] developed an automatic segmentation approach combined with post-processing tasks to identify the Fourier transform segmentation result in various phases of the CAD system, thus improving its performance, and this is compared with other state-of-the-art methods. These segmentation results help to determine the most effective feature extraction method and indicate how the morphology and textural information impact the accuracy of BC detection and classification.

Conventional extraction methods usually focus on obtaining low-level features of images; thus, prior information is needed to assist in selecting important features, which requires visual inspection undertaken by humans. Hence, DL techniques are used to automatically extract high-level features from images. DL methods are utilized in [14] to analyze breast histology images in both supervised and unsupervised settings. These DL techniques are able to adapt to the multi-class task of BC histology image classification, and they address dataset image invariability through the use of transfer learning and augmentation methods.

The authors of [15] present a saliency detection approach for the automatic isolation of ROIs in breast histology images. Several neural networks are trained on breast histology images at different magnifications, and individual results are obtained from these magnifications. They then construct training samples that are combined to form saliency maps to be used for the whole slide image. A CNN uses the obtained saliency maps to identify and classify different diagnostic categories. Eventually, both outputs from the ROI detection and classification tasks are combined to form a diagnosis.

Automatic BC classification in histology images is key in CAD analysis; thus, analysis efficacy is extremely important for early diagnosis. Therefore, DL methods are preferred for early diagnosis due to the fact that they give faster and more significant results than existing conventional machine learning methods. A novel DL approach proposed in [16] utilizes augmentation methods to individually pre-process images, which are later used as inputs to a fully connected deep convolution neural network model. Individual inputs are then processed via attention modules in the dense blocks within the model to obtain key ROIs, and the residual block of the neural network is used for image classification.

Image stain separation techniques are used to prepare images and provide color consistencies for cell structures in medical images. From these operations, pathologists are able to better analyze the resultant images and evaluate cell and tissue structures for malignancy growth and spread. The creation of many identical stain separation tasks or image transformations may lead to a combined output for cancer detection.

The authors of [17] propose a composite dilated backbone (CDB) network that links together several similar backbones (key features), forming a combined connection (feature vector) using different image pre-processing transformations. These connections (feature vectors) transfer high-level characteristics to the leading backbone layers in the fully connected CDB network, resulting in feature maps that assist in object detection and classification tasks. This proposed method improves object detection by integrating low-level and high-level data from several backbones within the network.

Traditional image-based classification methods rely on feature extraction methods to solve specific problems, thus offering limited functionality. To resolve this issue, ref. [18] presents a deep learning method for the classification of breast biopsy histology images. Images are labeled individually according to the different classes provided by the dataset. The images are then pre-processed via stain normalization and augmented to increase the dataset size. Consequently, CNNs perform patch-wise classification on the images through feature maps within the model, and, eventually, the fully connected network integrates information for the whole image patch and provides the classification.

Early CAD systems supplement analyses conducted by specialists, and ref. [19] presents a DL approach that utilizes transfer learning for the classification of breast histology images. Individual images are initially pre-processed via stain normalization, followed by data augmentation to create additional images necessary for the training and testing of the pre-trained CNN models. These CNN models are pre-trained and their hyperparameters are tuned for classification tasks on the resultant augmented image dataset.

Conventional machine learning methods fail to classify nuclei in histology images due to their complex microscopic structures; hence, Ref. [20] proposes an end-to-end deep neural network that resolves this problem. Dataset images are pre-processed into batch samples that act as inputs to the stacked de-noising auto-encoder model for unsupervised feature learning. The resultant extracted features are fed as inputs for training, distinguishing nuclei from non-nucleus objects via the softmax classifier. Lastly, the model is fine-tuned and used to classify new input histology image batches.

Another approach that resolves the problem of complex microscopic cell structure classification is proposed by [21] for mitotic nuclei. Images are pre-processed and stain-normalized using the method in [22]. A masked recurrent convolution neural network (R-CNN) is then used to distinguish non-mitotic from mitotic regions [23], thus performing both segmentation and detection on breast histology images.

Additionally, Ref. [24] proposes a similar deep learning method that focuses on maintaining informative nucleus regions while discarding non-nucleus regions. Images are pre-processed via data augmentation methods to increase the size of the dataset; the image patches are then resized and fed to the Inception-v3 CNN to extract nucleus-related features. Lastly, transfer learning through the hyperparameter tuning of the model is applied to enable image classification on whole slide images (WSI) in the dataset.

An automatic BC detection method proposed in [25] utilizes a three-class CNN model to segment hybrid cell features from microscopic structures in histology images. The model proposed is trained and tested to segment informative nucleus objects that are sparsely located in histology images. The morphology and spatial features are extracted to assist nucleus segmentation via highlighting ROI objects and their edges. Textural features are extracted from the obtained color information in the histology images (H&E). These outputs are fused via a relief-based feature section method [26], enabling the support vector machine to classify tumors as benign or malignant.

A belief theory-based classifier fusion strategy proposed in [27] is used for breast tumor classification. Images are pre-processed via the H&E staining procedure and then normalized using the [22] method to remove color inconsistencies and highlight nucleus regions. Additionally, a filter-based [28] blob detection method is used to identify nucleus objects, which are then extracted as nucleus patches. Consequently, these nucleus patches are fed into a pre-trained model consisting of fully connected CNNs that extract necessary features as vectors, which are finally classified via a support vector machine (SVM) as benign or malignant tumors.

The authors of [29] propose a deep learning-based technique for nucleus segmentation that targets microscopic feature information. Images are pre-processed via filtering and smoothing to improve the image quality and brightness, and morphology operations are further used to produce initial binary nucleus masks. These binary masks assist in extracting super-pixels from the original images, thus generating RGB image patches. These image patches are augmented and used as inputs to a CNN for training and testing purposes. The CNN outputs obtained are those of the extracted feature vectors, which are used segment nuclei in the ROIs of cancer histology images.

The paucity of medical images hinders accurate analysis, thus rendering the early diagnosis of BC difficult. The introduction of the U-Net model has successfully addressed the scarcity of images through its ability to produce high-accuracy evaluation scores with few input images. The authors of [30] present an improved U-Net model that uses EfficientNet as the main architecture for breast tumor cell segmentation and a multi-organ transfer learning method for the segmentation of the nuclei of breast tumor cells.

Organ cell images are obtained from different sources to form one dataset and their nuclei boundaries are annotated. The dataset is split into a train and test set, with the breast tumor cells forming the test set. The EfficientNet model is then used as an encoder input to the U-Net architecture. It processes the train set images using the squeeze-and-excitation components for optimization [31]. Each EfficientNet model (B0–B7) has distinct parameters, including the the image size, scale, resolution, and network depth, while the overall architecture remains the same. This leads to the uniform scaling of the network, targeting its width, its depth, and the resolution of the input images. Consequently, the U-Net decoder block, which is connected to the EfficientNet encoder block through a series of dense CNN blocks, utilizes transfer learning to segment the nucleus objects of BC cells via upsampling.

CAD systems offer decision support to medics and have resolved issues with conventional manual methods, such as the computational power and memory capacity, facilitating the processing of BC histology images. The authors in [32] present a review of various conventional and DL methods developed to analyze BC histology images, considering the characteristics of histology images, ROIs, image processing techniques, segmentation and classification methods, analysis challenges, and future work.

Due to the microscopic nature of tissue structures, it is necessary to topographically segment and analyze nucleus objects due to several challenges, such as improper image pre-processing, the complexity of tissue structures, overlapping cells, uneven color distribution caused by the staining procedure, and scanning equipment irregularities [33].

In [34], a fully convolutional deep neural architecture is utilized to segment nucleus cells. This technique uses image color normalization and standardization to highlight and balance the brightness of nucleus features and tissue structure regions, respectively. The Link-Net architecture is used to encode and decode images for nucleus segmentation.

The hematoxylin and eosin (H&E) image staining procedure, as presented in [35], normalizes and highlights the nucleus and tissue regions, respectively. The image contrast is adjusted to highlight and extract the nucleus from the image background and other regions. Watershed segmentation is performed on the extracted nucleus region to remove overlapping and unrelated objects not captured by the previous contrast adjustment step. A novel architecture proposed in [36] excludes intersecting pixels of overlapped nucleus objects to form a new foreground image. Morphological operations are applied to the new image to ascertain edges, nuclei are labeled using markers, and the final result is segmented via a watershed.

Another challenge when dealing with nucleus objects is the overlapping nuclei in histology images, and various methods have been proposed to resolve this. The authors in [37] propose a solution to isolate positively stained nucleus objects from a tissue image in an immunohistochemistry (IHC) sample. They pre-process the images using color deconvolution to separate the different components, namely positive nuclei (brown color) and negative nuclei (blue color), in the image. and a gray-level co-occurrence matrix (GLCM) is used for texture extraction. A deep learning approach, namely IHC-Net, is introduced to segment positive and negative objects and remove the non-diagnostic regions (including overlapping objects) from the image. Consequently, a proportion score is calculated as the number of positive nuclei to the number of positive and negative nuclei.

The effective early diagnosis of BC requires the precise identification of cancerous lesion boundaries/edges. This remains a difficult task, particularly for non-uniform regions in images. The authors in [38] propose the active contours method, which utilizes a local and global fitted function to extract unknown boundaries for regions of interest from non-uniform images. This method disregards false contours and a bias field is introduced for both the global and local fitted models to ensure the separation of the contours within an image.

BC tumors have a high risk of metastasizing and thus spreading to other parts of the body. The early detection of such metastatic regions can be achieved by predicting these tumors’ growth rates in histology images. The authors in [39] propose the novel Morpho-Contour Exponential Estimation (MoCEE) method, which utilizes an enhanced mask region-based convolution neural network (R-CNN) combined with active contours to ensure accurate BC tumor segmentation in magnetic resonance imaging (MRI) datasets.

This method pre-processes images via morphology operations to reduce noise and highlight salient lesion features and regions in the image. These salient features include texture and intensity variations and tumor boundaries/edges, while the ROIs include regions with unclear boundaries, indicating the presence of cancer or metastasis. These morphological features are then integrated into the masked RNN via an active contour function based on the image properties to allow further enhanced segmentation.

The active contour function primarily segments the breast tumor and analyzes the distance between two random points in the segmented region to determine the growth rate. The gradient boosting and exponential models are then used to predict tumor growth from the derived features via the iterative learning of feature vectors for better prognosis. The gradient boosting approach iteratively predicts the tumor size to approximate the true growth rate.

The problem of the inhomogeneity of edges and image boundaries also affects the segmentation and detection of BC in histology images, thus hindering the classification task. Additionally, there are challenges related to image objects, namely tiny objects referred to as blobs, which include image noise, low image quality/resolutions, and intersecting blobs.

To overcome these challenges, ref. [40] presents a joint deep learning and Hessian analysis small blob detection method for 3D images. This method utilizes the Difference of Gaussian (DoG) to pre-process images via smoothing and noise reduction. Hessian analysis then identifies potential blob candidates given a set local convexity threshold.

Subsequently, a trained U-Net that comprises fully convolutional layers, with the output size dependent on the input size, is utilized to obtain a probability map that captures potential blob locations. Lastly, the combination of the blob candidates and the probability map forms joint constraints to identify the true blobs in the image. The joint constraint highlights blob spots via a probability threshold to remove image noise. It also addresses the challenge of small blob detection by splitting intersecting objects not captured by the probability threshold.

The authors in [41] propose a blob detection approach for dental caries. They pre-process images via augmentation, thresholding, and segmentation via morphology operations. Consequently, dark and bright spot features extracted in the segmentation step are combined to form a carious lesion mask. The Laplacian of Gaussian (LoG) approach influences the detection of the carious lesion mask, thus assisting further edge segmentation via active contours to detect tooth edges/boundaries. A convexity threshold is used to identify blob candidates and eventually highlight carious dental blobs.

Medical imaging biomarkers are vital for the early diagnosis of breast cancer. Object identification/detection and segmentation are essential in improving the efficacy of biomarker identification. However, it can be challenging to detect microscopic objects/blobs due to noise, the image resolution, and intersecting objects, thus resulting in ROIs with significant false positives. Hence, Ref. [42] proposes a U-Net deep learning model with denoising capabilities for small blob detection. The U-Net model processes the provided dataset, and a probability map comprising potential blob locations is obtained. Lastly, Otsu thresholding is used to optimally identify blob candidates, thus addressing undersegmentation.

Other recent works that have used blob detection include [43]. In the cited work, the detection process introduced errors, leading to false positive and false negative results. To overcome this, their method introduces an effective image pre-processing technique to remove artifacts and segment the pectoral muscle from breast mammograms.

The authors in [44] also propose early image pre-processing to deal with false positive results, false negatives, and blob detection in the segmentation step.

This study utilizes a blob detection approach similar to [41]. Images are pre-processed and normalized via augmentation and stain normalization, respectively. Morphology operations are used for image enhancement, the CCA method is used to group objects with similar characteristics, and the active contours method highlights ROI edges/boundaries. A deep learning RNN uses this obtained information to automatically resolve undersegmentation from the previous step and optimally identify BC lesions (blobs) in breast histology images. The entire process is shown in Figure 1 and is described in detail in the following section.

## 3. Methods and Techniques

The proposed approach achieves the pre-processing, segmentation, and detection of BC in unsupervised histology images through the following steps.

### 3.1. Dataset Preparation and Pre-Processing

This study uses 24 unsupervised BC histology images from the publicly available Kaggle dataset repository.

Data are key for any neural network model to learn and deduce useful information from the data provided [45]. Recent artificial intelligence research has been heavily reliant on deep learning algorithms. These algorithms outperform conventional machine learning methods and thus rely on large datasets being available for model training. Hence, data scarcity in this case is addressed by applying a suitable method, namely data augmentation, which aims at increasing the dataset size, as large datasets are not publicly available, which is often the case with medical images.

#### 3.1.1. Data Augmentation

Data augmentation artificially creates additional data, which are used to train DL models, resulting in performance improvements when tested/validated on a separate unlabeled dataset. The authors in [46] present a study where data augmentation was utilized on medical images to train deep learning models. The review provides insights into these techniques and supports the validation of the resultant models.

The scarcity of publicly available datasets also leads to issues such as data bias, inaccurate results, and overfitting, but data augmentation resolves these issues. Data augmentation techniques improve the performance of the deep learning-based diagnosis of medical conditions in different organs, namely the breast, lung, brain, and eyes, via different imaging modes, such as mammography, computed tomography (CT), and magnetic resonance imaging (MRI), as examined in [47].

Data augmentation also entails artificially transforming existing images in a dataset by rotation, scaling, cropping, flipping, and height and width shifts to create more images. Augmentation is preferable based on its significant effectiveness in training different deep learning models [48]. Further, it assists in solving data scarcity issues by increasing the size and variety of images in datasets, which useful for the training of models without collecting new samples [49], and this increase in the dataset size assists in maintaining the image quality [18].

In this study, data augmentation methods such as rotation, scaling, and height and width shifts are used to increase the dataset size from 24 unsupervised BC histology images to 27,249 images.

#### 3.1.2. Data Stain Normalization

The emergence of medical imaging has led to advanced CAD systems and AI technologies that assist in digital pathology. The examination of tissue samples in medical images is commonly used to diagnose cancerous diseases, but the analysis of histology images is not always accurate. During the preparation and pre-processing stages, images exhibit various distortions and inconsistencies. These inconsistencies lessen the accuracy of computer-aided diagnosis, thus affecting pathologists’ diagnoses. Therefore, an effective stain normalization method is used to standardize and minimize color inconsistencies and variations in histology images. The authors of [50] review different stain normalization techniques, highlighting the main methodologies, contributions, strengths, and weaknesses, and rank them according to selected performance and accuracy scores.

In this study, color inconsistencies and variations are attributed to the H&E staining procedure, which highlights microscopic nucleic features, tissue structures, and image transformations resulting from data augmentation. Laboratory slide preparation, examination, analysis, and the digitalization of scanning samples are other factors that lead to image variations [51]. These factors negatively impact the training and testing of neural networks. Consequently, this study utilizes the Macenko et al. [22] stain normalization technique for BC images to separate nucleus features from tissue structures. The images in the dataset are first converted from the BGR to the RGB color space to enable smooth stain normalization.

**Macenko stain normalization**: This is used to prepare tissue slides. Image colors are converted to their optical density (OD) equivalents via a simple logarithmic transformation, as shown below:$$OD=-\log_{10}\left(I\right)$$
with *I* as the RGB color vector and individual components normalized to [0, 1].

A value β is used as a threshold value to remove data with a higher OD intensity.

Single value decomposition (SVD) is applied to optical density tuples to create a plane. The plane corresponds to the largest singular values. OD-transformed pixels are then projected onto the plane to determine the angle at each point related to the first SVD direction. The color space transformation is applied to the original BC histology image. An image histogram is stretched such that the range covers the lower (100−β)% of the data.

Minimum and maximum vectors are calculated and projected back to the OD space. The hematoxylin stain corresponds to the minimum vector, while the eosin stain is the maximum vector. Stain concentrations are determined to form a matrix representing the RGB channels and OD intensities, respectively. This study sets the values of α and β at 1 and 0.15, respectively. Figure 2 shows the original, augmented, normalized H&E, normalized H, and normalized E breast cancer histology images, respectively. After this stain normalization process, our proposed approach focuses on the normalized H image (image with only nucleus objects), having isolated it from the greater normalized H&E image set.

### 3.2. Image Enhancement

Dataset image enhancement improves the brightness, contrast, and scaling to compensate for the non-uniformity of image illumination. This study utilizes thresholding, morphology operations, and a distance transform to enhance the images in the dataset.

#### 3.2.1. Thresholding

Binary thresholding is used to capture the outline of a BC ROI in the normalized histology image/images and is shown in Figure 3.

#### 3.2.2. Morphology Operations

These include dilation and erosion operations to remove noise, remove overlapping edges, and extract certain regions from the BC histology images. Opening and closing operations are used to distinguish between the background and the foreground in an image. These regions are separated by diminishing and accentuating image pixels and edges. These operations also highlight the unknown area between the background and the foreground. Figure 4 shows images resulting from morphology operations.

#### 3.2.3. Distance Transform

This isolates nucleus objects in the image by locating the foreground and deleting remaining ROIs. It also highlights and emphasizes the foreground objects and background of the BC histology image. Figure 5 shows the image after distance transformation.

### 3.3. Segmentation

Nucleus segmentation is handled by the connected components analysis method and the active contours segmentation method.

#### 3.3.1. Connected Components Analysis

The CCA method combines pixel components with similar neighborhood properties. Additionally, image pixels are grouped into connected darker and brighter regions. Darker regions form the background, while brighter pixels form the foreground. The connected components analysis (CCA) method extracts these ROIs as binary masks from BC histology images. These binary mask ROIs are shown in Figure 6.

The detection of darker and brighter regions is achieved by the Laplacian of Gaussian approach, using a convolutional kernel of the form(1)$$LoG=\frac{x^{2}+y^{2}-2\sigma^{2}}{\sigma^{4}}e^{-\frac{x^{2}=y^{2}}{2\sigma^{2}}}$$
such that σ is the kernel width.

#### 3.3.2. Active Contours Segmentation

Active contours have been widely used in segmenting medical images, especially in computer vision tasks, to describe the boundary shapes in an image. They are widely utilized to resolve cases where the approximate shape of a boundary/edge is unknown. The active contours model adapts and evolves according to image color variations, enabling the matching and tracking of object boundaries/edges. The active contours method also accentuates elusive boundary (contours) shapes by ignoring missing/inhomogeneous boundary/edge information.

The authors in [52] propose an active contours-based model (snake) to detect the boundaries of objects from deformed initial contours. The deformed contours use an energy function that decreases when the snake perfectly fits the object boundary in an image. With an increased number of objects in the medical image, the snake approach experiences difficulties in segmenting the image. Therefore, the numerical step-by-step procedures proposed in methods used in [53,54] are used to detect topology changes automatically in the image.

The resultant binary-masked images from the previous connected components analysis are then used for contour detection. Multiple boundaries of objects are detected, as shown below.

Let $C\left(p,t\right):0,1\to R^{2}$ denote a family of curves resulting from the motion $C_{0}\left(P\right)$ directed towards inward Euclidean vector *N*. Let *I* denote the image where the object boundaries are to be identified and detected. We assume that the plane evolution of the curve is given by(2)$$\begin{array}{c} dCdt=g(I)[k+v]N,\\ C\left(p,t=0\right)=C_{0}\left(initialcurve\right), \end{array}$$
where *v* is a constant, *k* is the local curvature, *N* is the unit vector normal to the curve, and g(I) is a factor related to the image content.

We assume that the deforming curve C(p,t) is the zero value of a function ⋃, i.e., C(p,t) is a set of points (x,y,t) given U(x,y,t)=0. Given Equation (Equation 2) and the derivative of U(x,y,t)=0 with respect to space and time, the deformation of C(p,t) is given by the deformation of surface ⋃(x,y,t), whose evolution is given by(3)$$\begin{array}{c} dU(x,y,t)dt=g(I)dvi\nabla ⋃|⋃|+v|\nabla ⋃|,\\ ⋃\left(x,y,t=0\right)={⋃}_{0}\left(x,y\right), \end{array}$$

Here, |∇⋃| denotes the magnitude of the gradient, dvi denotes the divergence operator, and ${⋃}_{0}$ is the level set representation of $C_{0}$.

Consequently, the authors in [55,56] presented the concept of geodesic active contours, which results in a geometric model given by(4)$$\begin{array}{c} dU(x,y,t)dt=g(I)dvi\nabla ⋃|⋃|+v|\nabla ⋃|,+\nabla g\nabla ⋃\\ ⋃\left(x,y,t=0\right)={⋃}_{0}\left(x,y\right). \end{array}$$ From these results, a field is generated, ⋃(x,y,t), having null positions corresponding to active contour locations at any given evolution time. In these equations, *v* is a constant that constrains the active contours from either expanding or shrinking and is a function of the method used to draw the initial coarse contour. This constant is key within the model because it allows the initial curve to acquire a non-convex shape. dvi(∇⋃/|⋃|) denotes the curvature of the level set passing by a point and determines the regularizing effect of the model.

The function g(I) is utilized as a stopping factor in the evolving curve; specifically, the factor is small near an edge/boundary so as to stop the evolution when the contour moves close to the edge. The function g(I) is expressed as$$g\left(I\right)=1(1+|\nabla \left(I_{1}{|}^{p}\right),$$

Given the $I_{1}$ results from the low-pass Gaussian filtering of image I,p=1or2 and other expressions of g(I) can be used to monitor other features. The effect of ∇(g)∇⋃ is to capture the evolving contour as it moves towards an edge and push it back if it crosses the edge. Therefore, unlike the conventional snake method, the geometric active contours model is stable and handles topographical changes, namely splitting and merging, and is devoid of any computational problems. The active contours method extracts ROIs of interest, as shown in Figure 7.

### 3.4. Detection

Geometric features are the most important extraction features when detecting cancerous lesions in BC histology images. The previous active contours segmentation method extracts these geometric features, resulting in highlighted edge boundaries and the isolation of BC lesions from other ROIs in the image. Further, it is necessary to address oversegmentation brought about by the active contours segmentation failing to resolve the image edges’ inhomogeneity. This inhomogeneity leads to a high frequency of non-cancerous lesions (false positives) in the image.

Consequently, models experience longer processing times, leading to the poor performance of CAD systems. The authors in [41] adopt blob sensitivity parameters to select caries candidates and eliminate false positives from carious candidates. The present study uses a similar blob detection approach with high blobness values and low blobness values indicating BC and non-BC candidates, respectively. Therefore, the maximum and mean blobness values of cancerous candidates are used to eliminate false positives. The formulas are defined as(5)$$Blobness_{mean}\left(T\right)=\frac{\Sigma_{p\varepsilon T}Blobness\left(\lambda_{p}\right)}{N_{T}}$$(6)$$Blobness_{max}\left(T\right)=max_{p\varepsilon T}Blobness\left(\lambda_{p}\right)$$

The *T* refers to a cancerous candidate with $N_{T}$ voxels and *p* is a voxel belonging to a cancerous candidate. The size of the blobs is another feature extracted to aid in selecting BC candidates. The feature size is used since cancerous candidate detection based on Hessian analysis is sensitive to image intensity variations, which lead to false positives. The linear regression model [57] is applied to eliminate false positives within BC candidates. The cancerous selection function $L_{s}\left(T\right)$ is defined as(7)$$Z\left(T\right)=\beta_{0}+\sum_{i=1}^{N_{f}}\beta_{i}x_{i}$$(8)$$L_{s}\left(T\right)=\frac{1}{1+e^{-Z(T)}}$$
where $N_{f}$ is the number of features, $x_{i}$ is the feature value, $\beta_{0}$ is a constant coefficient, and $\beta_{i}$ is the corresponding coefficient estimated by the linear regression model. No threshold value is needed to determine the cancerous lesions after eliminating false positives, since geometric features have already been extracted via the active contours method in the segmentation step. Lastly, the remaining selected cancerous candidates are classified as “BC detected” and are shown in Figure 8.

The effect of the proposed blob detection method for unsupervised BC histology images was evaluated using a deep learning recurrent neural network model. The recurrent neural network architecture consisted of eight layers: one input dense layer, three hidden dense layers, one output dense layer, and three dropout layers. The cross-entropy was minimized using categorical cross-entropy with the Adam optimizer, with a learning rate of 0.0001, a batch size of 32, dropout of 0.2, and 30 epochs. We chose these hyperparameters based on iterative model experiments.

## 4. Results and Discussion

The experimental results of this study are based on a performance evaluation of the automatic blob detection method on unsupervised BC histology images. These experiments were carried out on 27,249 unsupervised BC histology images split into 20,436 training and 6813 testing set images. The data preparation, pre-processing, image enhancement, segmentation, and detection steps applied to the unsupervised BC dataset have been discussed in the preceding section.

Figure 1 shows an overview of the processing stages involved in the proposed detection method, namely pre-processing, image enhancement, segmentation, and detection. Consequently, the performance evaluation of the proposed approach is based on its automatic ability to detect cancerous lesions in unsupervised breast histology images, as shown in Figure 9.

These image results demonstrate the efficacy of the proposed method in detecting cancerous lesions in other histology images; it is not limited to breast histology images. Breast cancerous lesions are detected via their spread on tissues, signified by irregularities in the histology images. The H&E stain separation and normalization technique aids the separation of image pixels into hematoxylin (blue color), denoting nucleic features, and eosin (pink color), denoting tissue structures. Therefore, any other breast lesion on either lobules or ducts can be detected by separating nucleus objects from non-nucleus objects in the histology images, demonstrating the applicability of the proposed approach.

The proposed method uses various approaches to improve the efficiency in detecting BC in histology images. These approaches include data augmentation to deal with the small size of the available dataset and stain normalization to resolve color inconsistencies resulting from augmentation. Further, image enhancement is handled through thresholding and various morphology operations, namely dilation, erosion, and the distance transform. These enhancement techniques highlight the BC ROIs and remove noise and overlapping objects, thus isolating nucleic objects from non-nucleic objects.

The image results obtained from the enhancement phase are then segmented via the connected components analysis method and the active contours method. These methods are utilized to group components with similar characteristics into binary masks and resolve the ROI edges/boundaries’ inhomogeneity. Geometric features extracted from the resultant images are used to resolve any remaining blurred ROI edges/boundaries via the blob detection method. Consequently, this addresses oversegmentation and results in isolated BC lesions.

Our proposed technique achieved significant results in detecting cancerous lesions throughout the unsupervised breast histology image dataset. These results are attributed to various factors, namely dataset augmentation, stain normalization, image enhancement via morphology operations, and the mentioned segmentation methods.

Table 1 shows a comparison between our proposed approach and other state-of-the-art ROI detection-related techniques. From the literature reviewed, some of the methods discussed utilize image patches on individual images for the faster segmentation of whole slide images (WSI) and also aid data augmentation. Transfer learning has also been used in various methods in the literature to automatically extract feature vectors.

The DL methods discussed herein that use supervised images tend to produce high-performance results since the images are previously annotated and already pre-processed. There is also the use of pre-trained models such as VGG19 and ResNet to automatically segment, detect, and classify various ROIs. The preferred method in dealing with color inconsistencies is the one in [22] since it targets the specific nucleus ROIs.

For those methods discussed herein that do not utilize stain normalization but have produced significant results, emphasis is placed on the segmentation/detection methods applied to extract the necessary features. Most methods dealing with unsupervised images tend to focus on image pre-processing and stain normalization to assist in resolving data scarcity issues and color irregularities, respectively. Image enhancement techniques also play a pivotal role in highlighting and isolating important image features; thus, the utilization of morphology operations has been discussed in the literature. Segmentation and detection methods that use the obtained results from image enhancement stages tend to produce significant results due to focused image processing.

Figure 10 shows the model performance before weight regularization techniques are applied, thus exhibiting the presence of overfitting.

Figure 11 visually shows the model’s performance on unsupervised BC histology images—specifically, the detection of BC after fine-tuning its parameters. Weight regularization techniques, namely dropout and early stopping, were applied to deal with overfitting.

### Limitations

Oversegmentation and the remaining edge inhomogeneity in the active contours method used herein subsequently affect blob candidate selection. Moreover, most segmentation/detection methods tend to perform segmentation/detection on whole slide images (WSI) rather than image patches, thus increasing the model’s computational time.

Additionally, there is reluctance among pathologists to invest in CAD systems due to the large number of false positive results. Therefore, it is necessary to introduce deep learning neural networks within CAD systems to assist in the early diagnosis and treatment of breast cancer. Consequently, the proposed automatic blob detection method for cancerous lesions in human breast histology images can be used to isolate nucleus objects from other tissue structures in an image dataset, leading to faster model computability.

## 5. Conclusions

Recently, we have seen the increased utilization of CAD systems for the analysis of medical images, including breast histology images, and studies of how they can facilitate and assist in the early diagnosis of cancerous lesions, as well as segmentation and detection. Segmentation and detection are key tasks that aid imaging analysts in obtaining important information from medical images, including histology images.

Several computer-aided diagnosis systems have emerged that aid the extraction of ROIs to identify objects with similar features and characteristics in images for exploration purposes. These systems allow medical practitioners to interpret medical images and thus improve the efficiency of diagnosis and treatment tasks. Segmentation and detection tasks have been improved with the introduction of these automatic diagnostic systems.

This study has clearly shown the importance of prior image processing in effectively detecting cancerous lesions in breast histology images. The proposed approach has been systematically broken down into stages, namely pre-processing, image enhancement, segmentation, and detection. The preparation of the dataset involves resolving the data scarcity via data augmentation, while stain normalization addresses color inconsistencies resulting from augmentation.

Thresholding and morphology operations are used to remove noise and enhance unclear features in the resultant images, respectively. The segmentation methods used in this study specifically target nucleus regions in histology images by grouping components with similar characteristics, removing overlapping and non-nucleus objects, and resolving edge boundary inhomogeneity while topographically distinguishing between different image regions, namely nucleus ROIs, the foreground, and the background.

The outcome of the entire proposed technique is the development of a deep learning recurrent neural network that automatically detects cancerous lesions in unsupervised breast histology images. The neural network evaluates the detection task effectively and produces significant results. From the literature discussed and the results obtained with the proposed method, image pre-processing and image enhancement techniques, particularly augmentation and stain normalization, are crucial to the segmentation and detection task. These methods play a pivotal role in image feature extraction and in the detection of cancerous lesions in histology images, in turn improving the overall performance and model computability.

## 6. Future Work

CAD systems are necessary and offer a faster and better prognosis to assist in early treatment. These systems offer an alternative solution for medical practitioners when detecting BC lesions in histology images.

This study provides a valuable solution that enables models to correctly identify, separate, and detect cancerous lesions (nucleus objects and their edges) in BC histology images given the significant results obtained compared to other methods. The encouraging results of automatic blob detection offer insights and further avenues for exploration. Future work is not just limited to breast histology images and unsupervised images. Exploration areas include medical fields with publicly available datasets and the introduction of hybrid approaches. Additionally, weight regularization methods and the fine-tuning of supervised and unsupervised models can be explored to assist in the improvement of the model’s efficacy and reduce overfitting.

These perspectives are discussed in detail below.

**Data availability and integrity**. Most deep learning approaches require huge volumes of data to achieve meaningful performance results. Therefore, publicly available image datasets are necessary, especially histology image datasets, to assist deep learning.**Regularization methods**. These are needed to improve the performance of models. This can be achieved through model hyperparameter tuning, such as optimizing the learning rates, dropout, loss functions, activation functions, and early stopping methods.**Hybrid image processing/model approaches**. Combining various/several image processing methods or model architectures, it would be possible to form a hybrid method that improves the overall evaluation performance. This combination can occur at any step in the model, such as pre-processing, combining various attributes of different models to form one that will enhance the training, extraction, detection, and classification tasks. Additionally, future work could expand, explore, and diagnose other human and animal diseases through image datasets, moving beyond BC histology images.

## Figures and Tables

**Figure 1 bioengineering-12-00364-f001:**
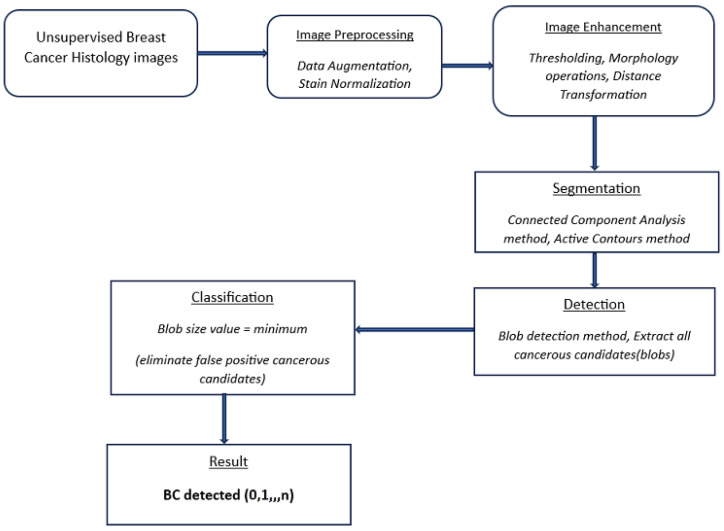
Flow diagram of the proposed detection method. Cancer histology images.

**Figure 2 bioengineering-12-00364-f002:**
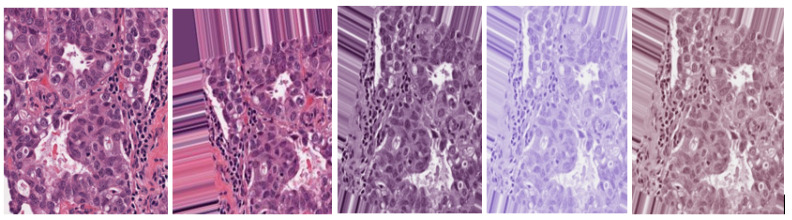
Original H&E image, augmented H&E image, normalized H&E image, normalized H image, and normalized E image, respectively.

**Figure 3 bioengineering-12-00364-f003:**
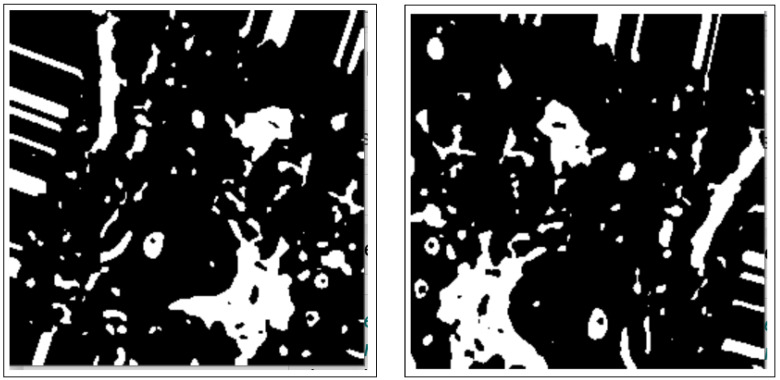
Images after binary thresholding.

**Figure 4 bioengineering-12-00364-f004:**
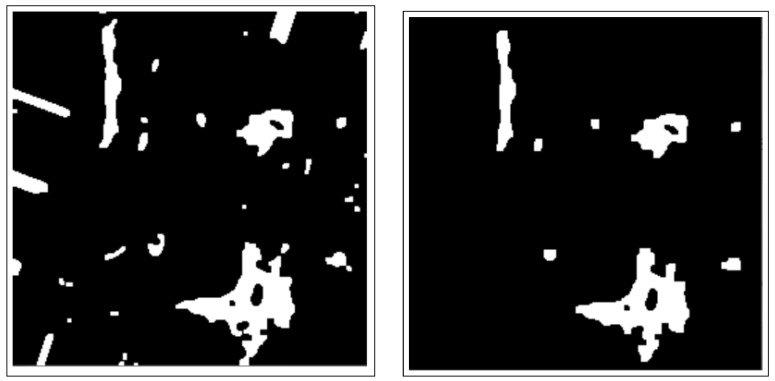
Images after morphology operations, namely erosion and dilation operations and opening operations (clearing borders), respectively.

**Figure 5 bioengineering-12-00364-f005:**
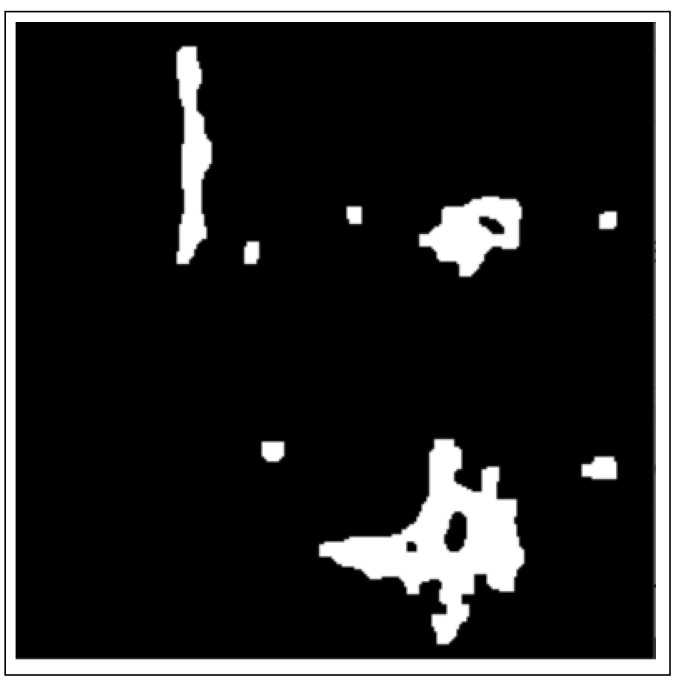
Image after distance transformation.

**Figure 6 bioengineering-12-00364-f006:**
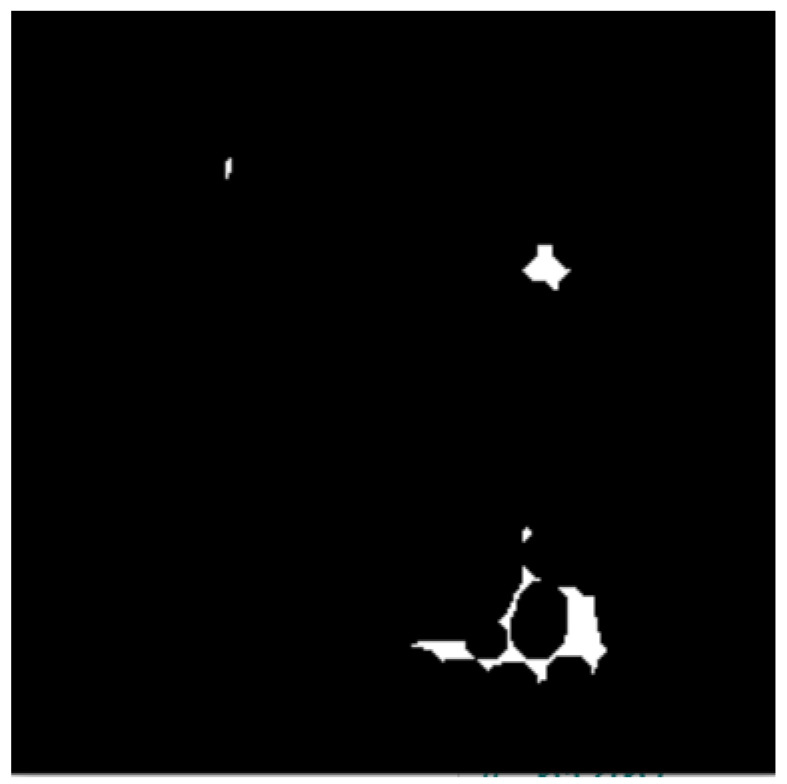
Image after connected components analysis.

**Figure 7 bioengineering-12-00364-f007:**
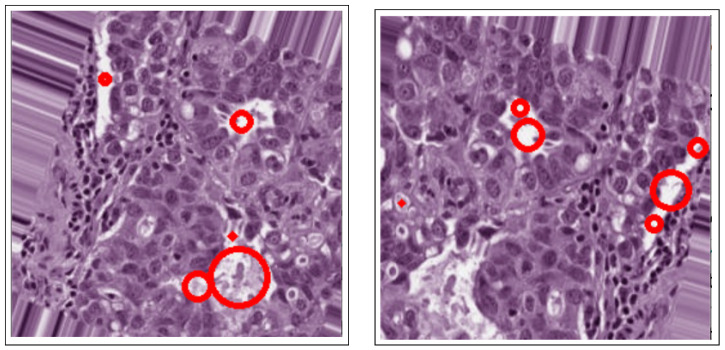
Active contours in histology images.

**Figure 8 bioengineering-12-00364-f008:**
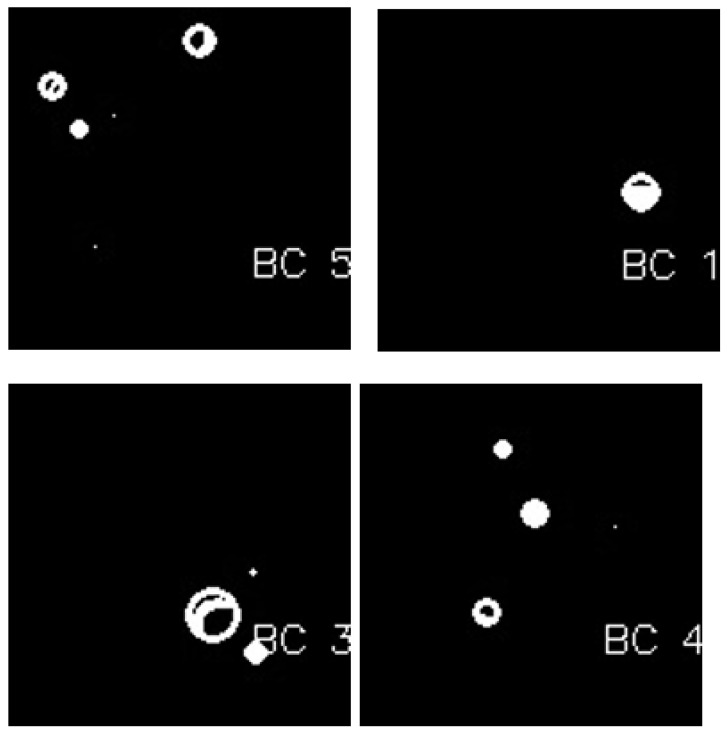
BC detection after blob detection in masked images.

**Figure 9 bioengineering-12-00364-f009:**
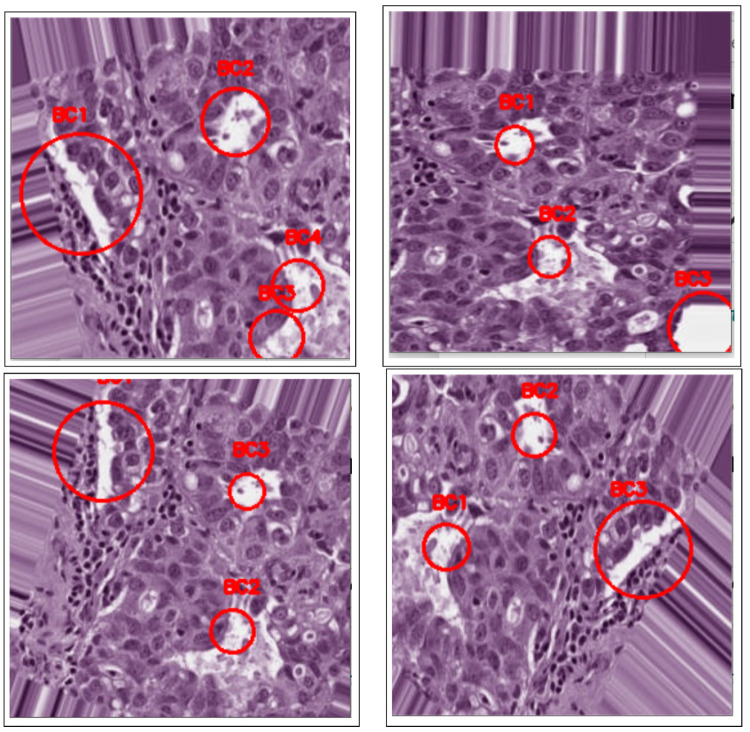
BC detection after blob detection in original images.

**Figure 10 bioengineering-12-00364-f010:**
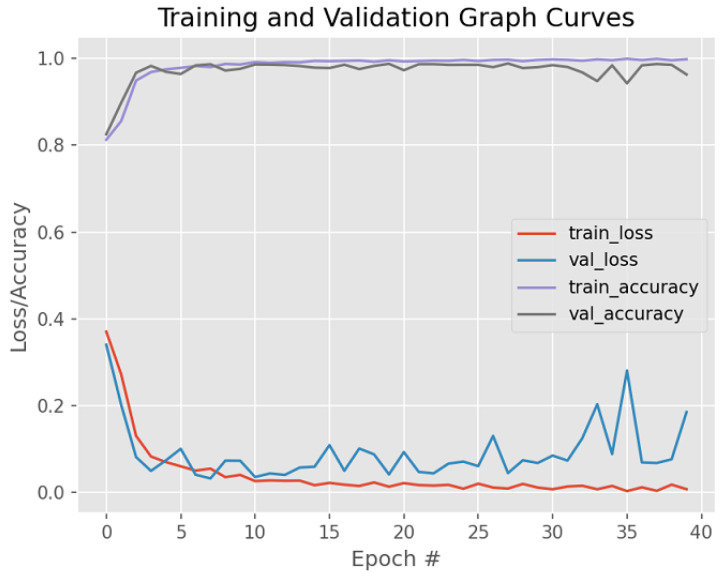
Model training/validation loss/accuracy graph curves before weight regularization techniques are applied.

**Figure 11 bioengineering-12-00364-f011:**
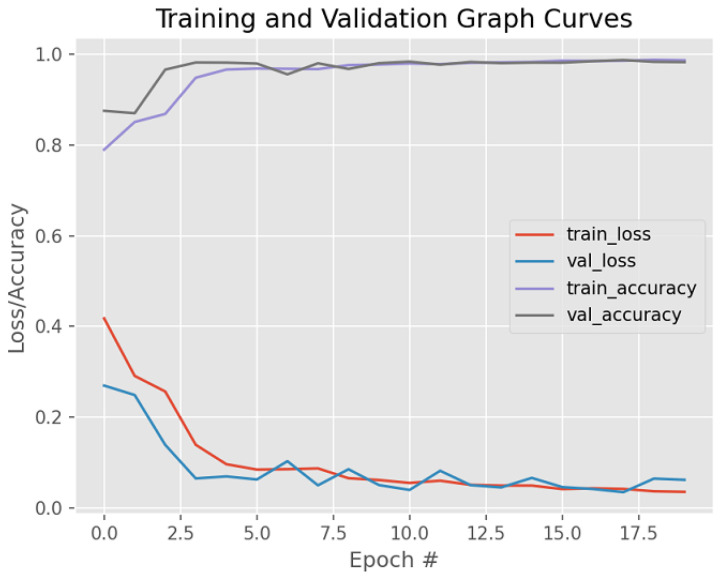
Model training/validation loss/accuracy graph curves after dropout and early stopping are applied.

**Table 1 bioengineering-12-00364-t001:** Performance evaluation and comparison with state-of-the-art methods.

Authors	Type of Image	Pre-Processing/Image Enhancement Methods	Segmentation Method	Detection Method	Accuracy
Reshma et al. [13]	Unsupervised images	Median filters, top- and bottom-hat filtering, grayscaling	Fourier transform	Speeded Up Robust Features (SURF) method	85.17%
Kiran et al. [10]	Unsupervised images	Color deconvolution, data augmentation	Binary thresholding, marker-controlled watershed algorithm	Dense Res-U-Net model	90.03%
Araujo et al. [18]	Supervised images	Macenko normalization, augmentation	CNN	Support vector machine (SVM)	95.6%
Isohail et al. [21]	Unsupervised images	Macenko normalization, mean standard deviation-based normalization	Masked R-CNN	Deep High Ensemble Mitotic Classifier (DHE-Mit-Classifier)	77%
Yu et al. [25]	Unsupervised images	Color deconvolution	Speeded Up Robust Features (SURF), gray-level co-occurrence matrix (GLCM), and local binary patterns (LBP)	Support vector machine (SVM)	96.7%
George et al. [27]	Supervised images	Macenko normalization	Laplacian of Gaussian (LoG)–blob detection algorithm	CNN (transfer learning)	96.3%
Sornapudi et al. [29]	Unsupervised images	Gaussian and median filters, linear transformation	Simple linear iterative clustering (SLIC) super-pixel algorithm	CNN (transfer learning)	95.70%
Veta et al. [33]	Unsupervised images	Color deconvolution, opening and closing morphology operations	Fast radial symmetry transform	Marker-controlled watershed	81.5%
Natarajan et al. [34]	Unsupervised images	Color and illumination normalization	LinkNet encoder–decoder architecture	LinkNet + Freeman chain coding (post-processing)	97.2%
Niaz [38]	Unsupervised images	None	Chan–Vese (CV) method, local binary fitting (LBF) method, local image fitting (LIF) method, variational level set with bias correction (VLSBC) method	Weighted length regularization by place (WLRP) method	98.39%
Xu et al. [40]	Unsupervised glomerulus images	Difference of Gradient (DoG), Hessian analysis	U-Net probability map	Hessian convexity map + U-Net probability map (blob detection)	96.3%
Majanga et al. [41]	Unsupervised dental images	Grayscaling, Gaussian blurring	Thresholding, erosion, dilation morphology, connected components analysis	Active contours, blob detection + convexity thresholding	97.0%
**Proposed Method**	**Unsupervised breast histology images**	**Augmentation, Macenko normalization, binary thresholding, erosion, dilation, opening and closing morphology operations, distance transformation**	**Connected components analysis (CCA) method, active contours (AC) method**	**Blob detection on (CCA+AC) outputs**	**98.82%**

## Data Availability

The data used to support the findings of this study can be obtained from the corresponding authors upon request.

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
