# Peer review of "Automatic Blob Detection Method for Cancerous Lesions in Unsupervised Breast Histology Images"

_bioengineering, 2025, doi:10.3390/bioengineering12040364_

Round 1

Reviewer 1 Report

Comments and Suggestions for Authors

Vincent Majanga et al. reported an interesting study about cancerous imaging detection. The topic was to some degree significance, and fell within the scope of Bioengineering. The manuscript could be reconsidered for publication after a Major Revision. Detailed comments:

1.       Please double-check the information of authors: (1) All 4 authors were listed as co-first authors; was this appropriate? (2) The institute information of 2~4 was missing.

2.       The Abstract needed to be rewritten. The current version should be shortened, and the significance of the findings should be added at the end of it.

3.       According to Introduction, there were 6 contributions of this paper. Please reformulate the contributions to 2~3 points; 6 points might sometimes be regarded as overestimation.

4.       A tabular or schematic summary of Section 2 should be supplemented. The current writing style was hard to follow.

5.       For the H&E images, there were many marginal regions seemed to be blur, especially Figure 7 and 9. How did this come from? Please be aware that manipulation of figure was not allowed.

6.       The Results & Discussion must be carefully revised. (1) Line 582 “Table.??” ? (2) Please divide this section into more subsections. Currently, only one subsection 4.1 existed, which was improper. (3) Implications for clinical application should be discussed. (4) The caption of Table 1 “My data” must be revised. Besides, the significant digits of accuracy should be unified.

7.       Section 5: It was recommended to divide this section into 2 sections: Future work and Conclusion.

8.       Please double-check the format of References.

Author Response

REVIEWER 1

 Concern 1:

     Please double-check the information of authors: (1) All 4 authors were listed as co-first authors; was this appropriate? (2) The institute information of 2~4 was missing.

Author’s Response:

The specific concerns have been addressed and are highlighted in blue in the paper.

Concern 2:

     The Abstract needed to be rewritten. The current version should be shortened, and the significance of the findings should be added at the end of it.

Authors’ Response:

 The specific concerns have been addressed and are highlighted in blue in the paper.

 Concern 3:

      According to Introduction, there were 6 contributions of this paper. Please reformulate the contributions to 2~3 points; 6 points might sometimes be regarded as overestimation.

Authors’ Response:

The specific concerns have been addressed and are highlighted in blue in the paper

 Concern 4:

  A tabular or schematic summary of Section 2 should be supplemented. The current writing style was hard to follow.

Authors’ Response:

 Section 2 has been formulated according to the various contributions mentioned in the introduction namely; augmentation methods, stain normalization, morphology operations, connected component analysis methods, active contours methods, and blob detection methods related to breast cancer which is our main topic of interest.

Concern 5:

For the H&E images, there were many marginal regions seemed to be blur, especially Figure 7 and 9. How did this come from? Please be aware that manipulation of figure was not allowed.

Authors’ Response:

Figures 7 and 9 show blobs on original augmented images thus the blurry nature of those images.

Concern 6:

  The Results & Discussion must be carefully revised. (1) Line 582 “Table.??” ? (2) Please divide this section into more subsections. Currently, only one subsection 4.1 existed, which was improper. (3) Implications for clinical application should be discussed. (4) The caption of Table 1 “My data” must be revised. Besides, the significant digits of accuracy should be unified..

Authors’ Response:

 The long array table was used in the paper due to the long details inside the table thus, it re-arranges the details inside the table differently as opposed to other forms of tables on overleaf. Raised concerns have been addressed and are highlighted in blue in the paper.

Concern 7:

Section 5: It was recommended to divide this section into 2 sections: Future work and Conclusion..

Authors’ Response:

 The raised concerns have been addressed and are highlighted in blue in the paper.

Concern 8:

Please double-check the format of References..

Authors’ Response:

 The format of referencing is the same all through the paper.

Reviewer 2 Report

Comments and Suggestions for Authors

In the paper “Automatic Blob Detection method for Cancerous Lesions in Unsupervised Breast Histology Images” is presented an approach to automatically detect hidden and inaccessible cancerous lesions in human breast histology images.    

The work presented by the authors is of interest, but there are some important points that need to be improved. My comments are as follows.

Although the architecture proposed by the authors appears to have the potential to improve the detection of hidden and inaccessible cancerous lesions in human breast histology images, the description of the approach lacks sufficient detail for the reader to understand it well.

There are some concepts that seem confusing. For example, what is meant by unsupervised images? There is a section called "Image Enhancement", where the authors start by saying that the goals are to improve images brightness, contrast and lighting, but in the subsections of this section, this is not demonstrated.

The main contributions list (lines 111-119) should not be formatted in italic.

The bibliographic review section must be reviewed in the following aspects:

- More recent work should be discussed. As far as I could understand, only 2 works from 2023 are referenced, with no reference to works published in 2024.

- This section should be reorganized to separate the analysis of proposals/works based on traditional techniques from more recent techniques.

Some statements need to be sustained by references. The sentences "The increased necessity for BC classification, ..." (lines 53-55) and "However, most segmentation..." (lines 67-68) are examples of these cases.

Figure 1 should be placed in the section, and near the text, where it's referred. Figure 2 is also misplaced. This figure visually exemplifies different techniques/methods but it is only referred in the subsection 3.1.2.

The techniques used for augmenting are basic, failing to consider more advanced techniques that have shown good results.

All equations must be numbered and the parameters/variables that compose them must be defined. Equations 3 and 4 formatting must be revised.

In the Results section, more results and respective analysis and discussion should be presented, and it is important to include visual results.

Table 1 label must be revised and its positioning corrected.

Comments on the Quality of English Language

The text must be carefully revised to correct some typing errors and improve sentence construction.

Author Response

REVIEWER 2

Concern 1

 The main contributions list (lines 111-119) should not be formatted in italic.

 Authors’ Response:

 The raised concerns have been addressed and are highlighted in blue in the paper.

Concern 2

 More recent work should be discussed. As far as I could understand, only 2 works from 2023 are referenced, with no reference to works published in 2024.

 Authors’ Response:

 The paper was written in 2024 that’s why the most recent work referenced is 2023, but in later work 2024 has been cited.

Concern 2

 This section should be reorganized to separate the analysis of proposals/works based on traditional techniques from more recent techniques..

 Authors’ Response:

 The entire paper is systematic in a way that the contributions mentioned in the introduction come out clearly in each step in the proposed method section. Tackling one issue after the other, thus the referencing is based on the specific work that is guiding to our study.

Concern 3

 Some statements need to be sustained by references. The sentences "The increased necessity for BC classification, ..." (lines 53-55) and "However, most segmentation..." (lines 67-68) are examples of these cases.

Authors’ Response:

 The raised concerns have been addressed and are highlighted in blue in the paper

Concern 4

 Figure 1 should be placed in the section, and near the text, where it's referred. Figure 2 is also misplaced. This figure visually exemplifies different techniques/methods but it is only referred in the subsection 3.1.2.

 Authors’ Response:

 Figure 1 has been placed where it should be, highlighting the various steps undertaken in section 3. Figure 2 is located right after the stain normalization method as it should be, the overleaf output brings it to a different position.

The raised concerns have been addressed and are highlighted in blue in the paper

Concern 5

 The techniques used for augmenting are basic, failing to consider more advanced techniques that have shown good results.

Authors’ Response:

The augmentation methods are standard augmentation methods used widely and have produced significant results herein. We will consider other methods in future work.

Concern 6

 All equations must be numbered and the parameters/variables that compose them must be defined. Equations 3 and 4 formatting must be revised..

Authors’ Response:

The raised concerns have been addressed and are highlighted in blue in the paper

Concern 7

 In the Results section, more results and respective analysis and discussion should be presented, and it is important to include visual results.

Authors’ Response:

Visual representation in form of graphs and images with blobs both on original and masked images have been displayed in our work. The raised concerns have been addressed and are highlighted in blue in the paper

Concern 8

 Table 1 label must be revised and its positioning corrected.

Authors’ Response:

The raised concern has been addressed and are highlighted in blue in the paper

Round 2

Reviewer 1 Report

Comments and Suggestions for Authors

Thanks for your revision.

Author Response

No reviewer comments provided.

Reviewer 2 Report

Comments and Suggestions for Authors

Figure 1 remains displaced. It is in section 2 and is referenced in section 1. Figure 2 remains displaced. The reason given by the authors is not satisfactory.

The literature review has not been updated. The authors give as a reason for this the fact that the paper was written in 2024. In my opinion, this is not a valid reason. The paper must be current and innovative at the time of submission. I reinforce my comment "This section should be reorganized to separate the analysis of proposals/works based on traditional techniques from more recent techniques", which was not addressed by the authors. Regarding my comment about the need to increase the number of visual results, the authors' response is not satisfactory. In table 1 the identification of the authors in the first column must be corrected. For example, writing [13] et al is not correct.   Comments on the Quality of English Language

The text should be carefully reviewed, as there are still sentences that need to be improved in writing.

Author Response

Manuscript ID: bioengineering-3454228

Title: Automatic Blob Detection method for Cancerous Lesions in Unsupervised Breast Histology Images

To: Bioengineering (ISSN 2306-5354)

 Re: Response to reviewers

 Dear Editor,

 Thank you for your response which allows addressing the reviewers’ comments. The revised version has been prepared, to incorporate the suggestions/comments of the reviewers. The reviewer's concern, response to that concern, and changes done in the manuscript are mentioned below.

Best regards,

Majanga et al.

REVIEWER 2

Concern 1

 Figure 1 remains displaced. It is in section 2 and is referenced in section 1. Figure 2 remains displaced. The reason given by the authors is not satisfactory.

Authors’ Response:

 The raised concerns have been addressed and are highlighted in blue in the paper.

Concern 2

 The literature review has not been updated. The authors give as a reason for this the fact that the paper was written in 2024. In my opinion, this is not a valid reason. The paper must be current and innovative at the time of submission. I reinforce my comment "This section should be reorganized to separate the analysis of proposals/works based on traditional techniques from more recent techniques", which was not addressed by the authors. Regarding my comment about the need to increase the number of visual results, the authors' response is not satisfactory. In table 1 the identification of the authors in the first column must be corrected. For example, writing [13] et al is not correct.

Authors’ Response:

 The literature review has been updated as requested and highlighted in blue. Our main focus is blob detection on breast cancer and thus not much has been published that we is related and could link with our work.  The way our work is organized is systematic as mentioned in the introduction part where specified our contributions towards the area of study. Each related works is linked to what specifically it is  addressing eg. Pre-processing, image enhancement, segmentation, detection etc. Visual representation is sufficient especially with our area of study which is detection, we have given images of detection on both the original images and on the final masked images. Raised concerns about the table have been addressed and highlighted in blue.

Round 3

Reviewer 2 Report

Comments and Suggestions for Authors

I have no further comments.